systems biology/mathematical modelling

cooperation, dynamic game, Stackelberg games, leaders in turn

**Author for correspondence:**
Pu-yan Nie
e-mail: pynie2013@163.com

# Players acting as leaders in turn improve cooperation

Pu-yan Nie[1], Chan Wang[1] and Ting Cui[2]

[1]Institute of Guangdong Economy and Social Development, School of Finance, and
[2]School of Accounting, Guangdong University of Finance and Economics (GDUFE), 510320 Guangzhou, People's Republic of China

P-yN, 0000-0002-6214-5980

Cooperation behaviour is an important topic in society as well as in the biological field, and many factors yield cooperation. Many social phenomena constitute Stackelberg games, but there is little literature on the relationship between Stackelberg games and cooperation. This article shows that in the repeated dynamic Stackelberg games, players acting as leaders in turn yields cooperation. Moreover, social welfare is improved correspondingly when players act as leaders in turn. Therefore, for dynamic Stackelberg games, this paper proposes that the institution of players acting as leaders in turn promotes cooperation.

## 1. Introduction

Many situations, including those in economics, society, management, the political and biological fields, are all hierarchical and these phenomena are modelled as static Stackelberg games (or as leader–follower games) [1]. Stackelberg games are initially proposed to model leader–follower situations in economics by Von Stackelberg [1]. In a static Stackelberg game, one player acts as the leader and other players play the followers' positions. When the leader makes decisions, he should consider the followers' strategies. The followers respond according to the leader's strategy. Following Stackelberg's classic model, there exist many types of extension. On the one hand, many papers have explored the theory of Stackelberg games. On the other hand, some authors developed the applications of Stackelberg games. Recently, based on many interesting social phenomena, Nie proposed a dynamic Stackelberg game with alternating leaders and analysed the properties of the game [2–4]. Nie [3] defined the dynamic Stackelberg game with players acting as leaders in turn as 'A dynamic Stackelberg game is called a dynamic Stackelberg game with players acting as leaders in turn, if all players act as leaders in turn in this game'. Actually, players acting as leaders in turn is very popular in society and has attracted many researchers' attention all over the world in many fields.

According to the theory of cooperation, many factors yield cooperation. For example, social diversity yields cooperation of

**Table 1.** Nomenclature.

| | |
|---|---|
| $I = \{1, 2\}$ | two players |
| $S = \{A, B\}$ | strategy space |
| $a_1(b_1)$ | payoffs of the first (second) player under strategy $(A, A)$ |
| $a_2(b_2)$ | payoffs of the first (second) player under strategy $(A, B)$ |
| $a_3(b_3)$ | payoffs of the first (second) player under strategy $(B, A)$ |
| $a_4(b_4)$ | payoffs of the first (second) player under strategy $(B, B)$ |

public goods [5]. Punishment is also closely related to cooperation and there is an ongoing debate on the relationship between punishment and cooperation. Some researchers thought that punishment also improves cooperation [6,7], while others proposed that costly punishment does not promote cooperation in society [8]. Nowak significantly proposed that five rules, including direct reciprocity, indirect reciprocity, spatial selection, multi-level selection and kin selection, all result in cooperation [9,10]. Nowak's work has attracted much attention in biology and society. Following Nowak's interesting five rules, an extensive body of literature has appeared in recent years to develop the theory of cooperation [11–13]. There are other factors that induce cooperation. For instance, zealots in a group also promote cooperation [14]. Further, institution also affects cooperation. In fact, it is difficult to achieve high levels of cooperation in a group of non-cooperative games. Through a laboratory experiment, high levels of cooperation appear if the institution is imposed on the group but not if development of the institution is left to the group to vote on [15]. There exist other interesting review papers about the evolution of cooperation [16,17].

Other factors also lead to cooperation in real life and the extant literature neglects the relationship between players' positions and cooperation except for Taylor et al. [18] and McNamara et al. [19] and references therein. Taylor et al. [18] argued that Stackelberg games improve cooperation compared with Nash games. This article aims to further explore the relationship between Stackelberg games and cooperation and we focus on a type of dynamic Stackelberg game in which players act as leaders in turn. There also exist some papers which introduced players without fixed positions. Nowak & Sigmund [20] initially introduced alternating Prisoners' Dilemma and Hauert et al. [21] developed the corresponding cooperation theory. Actually, Nowak & Sigmund [20] investigated special cases of the dynamic Stackelberg games with players acting as leaders in turn. In the alternating Prisoners' Dilemma, the strategy of the leader at some stage is exactly the follower's strategy at the previous stage. Diamani et al. [22] interestingly showed that the diversity emergency yields cooperation between cells.

This paper argues that for dynamic Stackelberg games, players acting as leaders in turn also leads to cooperation by comparing the profits of the players of classic dynamic Stackelberg games with those of the ones with players acting as leaders in turn. We thus propose a new type of cooperation rule based on many social phenomena. The nomenclature of this article is listed in table 1.

The rest of this paper is organized as follows: the model or method is presented in the next section. We present the payoff matrix at the static case. Results are analysed in §3. The model is considered both under classic dynamic Stackelberg situation and dynamic Stackelberg with leaders in turn. We compare them and conclude that leaders in turn improve the cooperation. Conclusions are remarked on in the final section.

## 2. Method

Here, we establish the model to discuss the relationship between cooperation and the players' positions. Assume there are two players denoted $I = \{1, 2\}$ and there are two strategies $S = \{A, B\}$ for the two players. The strategy spaces of two firms are identical. The payoff matrix is presented as follows.

When two firms launch strategy $(A, A)$, the first firm's payoff is $a_1$ and the second firm's payoff is $b_1$. The other notations are similarly defined. We always stipulate that $a_1 > a_3 > 0$ and $b_1 > b_2 > 0$ to guarantee the existence of the pure Nash equilibrium $(A, A)$ at the static state. We define the cooperation, in which two firms' payoffs are all promoted or a Pareto promotion.

We then analyse the above model in static cases, classic dynamic situation and dynamic cases with leaders in turn.

**Table 2.** Payoff matrix.

| | A | B |
|---|---|---|
| A | $a_1, b_1$ | $a_2, b_2$ |
| B | $a_3, b_3$ | $a_4, b_4$ |

## 2.1. Static equilibrium

Here we consider the above payoff matrix under the static Nash game. In this case, two plays simultaneously make decisions without information on the opponent's strategies.

(1) If $a_2 > a_4$ or $b_3 > b_4$, there exists the unique pure Nash equilibrium $(A, A)$.
(2) If $a_2 < a_4$ and $b_3 < b_4$, there exists two pure Nash equilibria $(A, A)$, $(B, B)$ and a mixed one

$$\left( \frac{b_4 - b_3}{b_4 - b_2 + b_1 - b_3} A + \frac{b_1 - b_2}{b_4 - b_2 + b_1 - b_3} B, \frac{a_4 - a_2}{a_1 - a_2 + a_4 - a_3} A + \frac{a_1 - a_3}{a_1 - a_2 + a_4 - a_3} B \right). \qquad (2.1)$$

To address the Stackelberg games with both the static cases and the repeated situations with the payoff matrix of table 2, we always assume that $a_1 > a_3$, $b_1 > b_2$, $a_2 < a_4$ and $b_3 < b_4$.

Under static Stackelberg situations and with the first player always acting as the leader, we have the following equilibrium: (1) If $a_1 > a_4$ (or $a_1 < a_4$), there exists the unique Stackelberg equilibrium: $(A, A)$ (or $(B, B)$). (2) If $a_1 = a_4$, there are two Stackelberg equilibria: $(A, A)$ and $(B, B)$.

Under static Stackelberg situations and with the second player throughout acting as the leader, we have the following equilibrium: (1) If $b_1 > b_4$ (or $b_1 < b_4$), there exists the unique Stackelberg equilibrium: $(A, A)$ (or $(B, B)$). (2) If $b_1 = b_4$, there are two Stackelberg equilibria: $(A, A)$ and $(B, B)$.

# 3. Results

This paper mainly addresses the infinitely repeated Stackelberg game with $a_1 > a_3$, $b_1 > b_2$, $a_2 < a_4$ and $b_3 < b_4$. At each stage, one player acts as the leader and the other is the follower. To simplify, we throughout assume that the discount factor is $\delta \in (0, 1)$.

## 3.1. Classic repeated dynamic Stackelberg game

In the classic repeated dynamic Stackelberg game, one player always plays the leading position and the others always act as followers. If the first player always acts as the leader, the sub-game perfect equilibrium is

(1) If $a_1 > a_4$ (or $a_1 < a_4$), the first player always uses strategy $A$ (or $B$) and the second player correspondingly employs $A$ (or $B$). There exists a unique equilibrium trajectory: always $(A, A)$ (or always $(B, B)$).
(2) If $a_1 = a_4$, at each stage, the first player randomly employs the strategy $A$ or $B$. If the first player employs strategy $A$, the second player correspondingly uses $A$. Otherwise, the second player uses $B$.

If the second player always acts as the leader, the sub-game perfect equilibrium is presented as follows:

(1) If $b_1 > b_4$ (or $b_1 < b_4$), the second player always uses strategy $A$ (or $B$) and the first player correspondingly employs $A$ (or $B$). There exists a unique equilibrium trajectory: always $(A, A)$ (or always $(B, B)$).
(2) If $b_1 = b_4$, at each stage, the second player randomly employs strategy $A$ or $B$. If the second player employs strategy $A$, the first player correspondingly uses $A$. Otherwise, the first player uses $B$.

## 3.2. The repeated dynamic Stackelberg game with players acting as leaders in turn

In this case, two players act as leaders in turn. At each stage, when one player acts as the leader, the other acts as the follower. Here we address the repeated dynamic Stackelberg game with players acting as

**Table 3.** Payoff matrix for the example.

|   | A | B |
|---|---|---|
| A | 5, 1 | 0, 0 |
| B | 4, 4 | 1, 5 |

leaders in turn. Without loss of generality, we always assume that the first player acts as the leader at the first stage. The sub-game perfect equilibrium is presented as follows:

(1) If $a_1 > a_4$ and $b_1 > b_4$ (or $a_1 < a_4$ and $b_1 < b_4$) $(A, A)$ (or $(B, B)$) is always the unique equilibrium trajectory.
(2) If $a_1 > a_4$ and $b_1 < b_4$, $\{(A, A), (B, B),(A, A), (B, B),\ldots\}$, the equilibrium in which $(A, A)$ and $(B, B)$ are alternately employed is an equilibrium trajectory. Moreover, if $a_1 - a_3/(a_3 - a_4) < \delta$ and $b_4 - b_3/(b_3 - b_1) < \delta$, the following strategy trajectory constitutes an equilibrium: always employing $(B, A)$ if no player breaks off.

If one player does not employ $(A, B)$ at some stage, after this stage, the two players use $\{(A, A), (B, B), (A, A), (B, B),\ldots\}$. If the two players always employ $(B, A)$, the social welfare is improved. For the sub-game perfect equilibrium $\{(A, A), (B, B), (A, A), (B, B),\ldots\}$, the profits of the two players are represented by

$$\pi_1 = a_1(1 + \delta^2 + \delta^4 + \cdots) + a_4(\delta + \delta^3 + \delta^5 + \cdots) = \frac{a_1}{1 - \delta^2} + \frac{\delta a_4}{1 - \delta^2} \tag{3.1}$$

and

$$\pi_2 = b_1(1 + \delta^2 + \delta^4 + \cdots) + b_4(\delta + \delta^3 + \delta^5 + \cdots) = \frac{b_1}{1 - \delta^2} + \frac{\delta b_4}{1 - \delta^2}. \tag{3.2}$$

For the strategy of always employing $(B, A)$, the profits of the two players are

$$\bar{\pi}_1 = a_3(1 + \delta + \delta^2 + \delta^3 + \cdots) = \frac{a_3}{1 - \delta} \tag{3.3}$$

and

$$\bar{\pi}_2 = b_3(1 + \delta + \delta^2 + \delta^3 + \cdots) = \frac{b_3}{1 - \delta} \tag{3.4}$$

When $a_3(1 + \delta) > a_1 + a_4\delta$ and $b_3(1 + \delta) > b_1 + b_4\delta$, or under the hypothesis of $a_1 - a_3/(a_3 - a_4) < \delta$ and $b_4 - b_3/(b_3 - b_1) < \delta$, the strategy of always employing $(B, A)$ Pareto dominates the sub-game perfect equilibrium $\{(A, A), (B, B), (A, A), (B, B),\ldots\}$. Therefore, the strategy of always employing $(B, A)$ if no player breaks off is an equilibrium. In this way, leaders in turn improve the cooperation.

Here an example of a repeated dynamic Stackelberg game with players acting as leaders in turn is listed to illustrate the above cooperation.

*Example.* Here we present a dynamic Stackelberg game with two players acting as leaders in turn. The payoff matrix of the static game with two players is outlined as follows (table 3) and the notations are the same as in table 3.

There exist two pure Nash equilibria $(A, A)$, $(B, B)$ and a mixed one for the static game with table 3. Here we address the repeated dynamic Stackelberg game with players acting as leaders in turn. In the above game, $\{(A, A), (B, B), (A, A), (B, B),\ldots\}$ is a sub-game perfect equilibrium. Moreover, if $\delta > 1/3$, the strategy of always employing $(B, A)$ is a sub-game perfect equilibrium if both the two players follow this strategy. This example completely meets the above theoretic conditions.

## 4. Discussion

This paper develops the theory of cooperation and proposes a new rule to cooperate. By infinitely repeated Stackelberg games, under rational decisions, this paper interestingly argues that players acting as leaders in turn improve cooperation and the social welfare. Or the leaders in turn yield a Pareto promotion. Therefore, a new cooperation mechanism is proposed in this article.

In general, the leader in the static Stackelberg games owns first-mover advantage. In the repeated Stackelberg games with players acting as leaders in turn, this advantage is weakened because the leader has to consider their follower positions at the subsequent stages. Therefore, without complete first-move advantages, players acting as leaders in turn improves cooperation.

Compared with the existing literature, Taylor *et al.* [18] showed that Stackelberg games improve cooperation upon Nash games, while this paper highlights the formulations of the dynamic Stackelberg games and shows that players acting as leaders in turn in dynamic games also improves the cooperation.

Nowak & Sigmund [20] focused on the evolutionarily stable strategy for alternating Prisoners' Dilemma. Compared with the alternating Prisoners' Dilemma, this paper differs from Nowak & Sigmund [20] both in the model and in the highlights. Hauert *et al.* [21] also addressed the cooperation of alternating Prisoners' Dilemma and argued that alternating the two players leads to the Prisoners' Dilemma with cooperation. The conclusions in this paper are in good agreement with Hauert *et al.* [21], although the model in this paper (or the situation) is significantly different from Nowak & Sigmund [20] and Hauert *et al.* [21]. Furthermore, this paper stresses the dynamic leader–follower games, while Nowak & Sigmund [20] and Hauert *et al.* [21] all focused on the dynamic Nash games. It is interesting to combine the reality, including emergent behaviours [22], resource constraints [23–25], moral code [26] and so on [27,28].

Data accessibility. This article does not contain any additional data.

Authors' contributions. P.Y.N. conceived the project, developed and analysed the model and wrote the paper. C.W. and T.C. contributed to ideas, design, calculations and wrote drafts of the manuscript. All authors approved the final version to be published.

Competing interests. We declare we have no competing interests.

Funding. P.Y.N. is partially supported by Guangdong Social Science Foundation in Finance (GD18JRZ05), National Natural Science Foundation of PRC (71771057). C.W. is partially supported by Humanities and social sciences fund of the Ministry of Education (18YJC790156), Guangdong Social Science Foundation (GD17XYJ23), Guangdong Natural Science Foundation (2018A030310669), and Innovative Foundation (Humanities and Social Sciences) for Higher Education of Guangdong Province (2015WCXTD009, 2017WQNCX053), Major Applied Project for Universities of Guangdong (Social Science) (2017WZDXM011).

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
