## [Reviewer comments · Royal Society Open Science]

Review History

RSOS-190251.R0 (Original submission)

Review form: Reviewer 1

Is the manuscript scientifically sound in its present form?

Yes

Are the interpretations and conclusions justified by the results?

Yes

Is the language acceptable?

Yes

Is it clear how to access all supporting data?

Yes

Do you have any ethical concerns with this paper?

Yes

Have you any concerns about statistical analyses in this paper?

Yes

Recommendation?

Accept with minor revision (please list in comments)

Comments to the Author(s)

The paper is well modified according to the former comments. However, there are still some minor comments.

- 1) A table of variables should be added to make the readability of the paper.
- 2) The contribution of the paper is better to be mentioned both in the introduction section and the conclusion section.
- 3) Literature review is not comprehensive enough. Actually, some latest researches is ignored by authors, which should be added..
- 4) The language of the paper should be modified. There are some grammatical errors, vague expression and colloquial problems.
- 5) Normalized format is required for an academic paper. There are some nonstandard expression needs to be modified. In references, the format of 23-25 are different from others.

Review form: Reviewer 2

Is the manuscript scientifically sound in its present form?

Yes

Are the interpretations and conclusions justified by the results?

Yes

Is the language acceptable?

Yes

Is it clear how to access all supporting data?

Yes

Do you have any ethical concerns with this paper?

No

Have you any concerns about statistical analyses in this paper?

No

Recommendation?

Accept with minor revision (please list in comments)

Comments to the Author(s)

By establish a Stackelberg model, this paper shows players acting as leaders in turn yields corporation. However, I think this paper still has some room for improvement.

1. The formulas are not numbered.
2. I suggest adding a list of symbolic meanings
3. Can this paper analyze the equilibrium under the infinite strategy?
4. Some language expressions need to be improved.

Decision letter (RSOS-190251.R0)

14-May-2019

Dear Miss NIE

On behalf of the Editors, I am pleased to inform you that your Manuscript RSOS-190251 entitled "Players acting as leaders in turn improve cooperation" has been accepted for publication in Royal Society Open Science subject to minor revision in accordance with the referee suggestions. Please find the referees' comments at the end of this email.

The reviewers and handling editors have recommended publication, but also suggest some minor revisions to your manuscript. Therefore, I invite you to respond to the comments and revise your manuscript.

- Ethics statement

- Data accessibility

<http://datadryad.org/submit?journalID=RSOS&manu=RSOS-190251>

- Competing interests

- Authors' contributions

AB carried out the molecular lab work, participated in data analysis, carried out sequence alignments, participated in the design of the study and drafted the manuscript; CD carried out

the statistical analyses; EF collected field data; GH conceived of the study, designed the study, coordinated the study and helped draft the manuscript. All authors gave final approval for publication.

- Acknowledgements

- Funding statement

Because the schedule for publication is very tight, it is a condition of publication that you submit the revised version of your manuscript before 23-May-2019. Please note that the revision deadline will expire at 00.00am on this date. If you do not think you will be able to meet this date please let me know immediately.

- 1) A text file of the manuscript (tex, txt, rtf, docx or doc), references, tables (including captions) and figure captions. Do not upload a PDF as your "Main Document";
- 2) A separate electronic file of each figure (EPS or print-quality PDF preferred (either format should be produced directly from original creation package), or original software format);
- 3) Included a 100 word media summary of your paper when requested at submission. Please ensure you have entered correct contact details (email, institution and telephone) in your user account;
- 4) Included the raw data to support the claims made in your paper. You can either include your data as electronic supplementary material or upload to a repository and include the relevant doi within your manuscript. Make sure it is clear in your data accessibility statement how the data can be accessed;

5) All supplementary materials accompanying an accepted article will be treated as in their final form. Note that the Royal Society will neither edit nor typeset supplementary material and it will be hosted as provided. Please ensure that the supplementary material includes the paper details where possible (authors, article title, journal name).

on behalf of Dr Robert MacKay (Associate Editor) and Mark Chaplain (Subject Editor)
openscience@royalsociety.org

Associate Editor Comments to Author (Dr Robert MacKay):

Associate Editor: 1

Comments to the Author:

I recommend to accept the paper subject to satisfactory revision according to the comments of the two reviewers.

Reviewer comments to Author:

Reviewer: 1

Comments to the Author(s)

The paper is well modified according to the former comments. However, there are still some minor comments.

1) A table of variables should be added to make the readability of the paper.

- 2) The contribution of the paper is better to be mentioned both in the introduction section and the conclusion section.
- 3) Literature review is not comprehensive enough. Actually, some latest researches is ignored by authors, which should be added..
- 4) The language of the paper should be modified. There are some grammatical errors, vague expression and colloquial problems.
- 5) Normalized format is required for an academic paper. There are some nonstandard expression needs to be modified. In references, the format of 23-25 are different from others.

Reviewer: 2

Comments to the Author(s)

By establish a Stackelberg model, this paper shows players acting as leaders in turn yields corporation. However, I think this paper still has some room for improvement.

1. The formulas are not numbered.
2. I suggest adding a list of symbolic meanings
3. Can this paper analyze the equilibrium under the infinite strategy?
4. Some language expressions need to be improved.

Author's Response to Decision Letter for (RSOS-190251.R0)

See Appendix A.

Decision letter (RSOS-190251.R1)

10-Jun-2019

Dear Professor NIE,

I am pleased to inform you that your manuscript entitled "Players acting as leaders in turn improve cooperation" is now accepted for publication in Royal Society Open Science.

on behalf of Dr Robert MacKay (Associate Editor) and Mark Chaplain (Subject Editor)
openscience@royalsociety.org

Associate Editor Comments to Author (Dr Robert MacKay):

The revised paper looks good. It is an interesting topic and could have wide application.

Appendix A

Dear Prof.,

Sincere thanks to you for your suggestions. All suggestions are fixed one by one.

Associate Editor Comments to Author (Dr Robert MacKay):

Associate Editor: 1

Comments to the Author:

I recommend to accept the paper subject to satisfactory revision according to the comments of the two reviewers.

REPLY: Sincere thanks to you for your valuable suggestions. Comments of two reviewers are fully considered in the revision version.

Reviewer comments to Author:

Reviewer: 1

Comments to the Author(s)

The paper is well modified according to the former comments. However, there are still some minor comments.

1) A table of variables should be added to make the readability of the paper.

REPLY: In the modified version, a table of symbols is presented as follows

Table 1 Nomenclature.

$I = \{1, 2\}$	Two players
$S = \{A, B\}$	Strategy space
$a_1(b_1)$	Payoffs of the first (second) player under strategy (A, A)
$a_2(b_2)$	Payoffs of the first (second) player under strategy (A, B)
$a_3(b_3)$	Payoffs of the first (second) player under strategy (B, A)
$a_4(b_4)$	Payoffs of the first (second) player under strategy (B, B)

2) The contribution of the paper is better to be mentioned both in the introduction section and the conclusion section.

REPLY: Sincere thanks to the anonymous review to attract authors attention. In the introduction part, we also stress that “We thus propose a new type of cooperation rule based on many social phenomena.” In the modified version of the conclusion section, we point out that “Therefore, a new cooperation mechanism is proposed in this article.

3) Literature review is not comprehensive enough. Actually, some latest researches is ignored by authors, which should be added..

REPLY: In the modified version, references 26-28 are added and these three papers are remarked.

4) The language of the paper should be modified. There are some grammatical errors, vague expression and colloquial problems.

REPLY: In the modified version, this article is carefully checked and some grammatical errors are modified.

5) Normalized format is required for an academic paper. There are some nonstandard expression needs to be modified. In references, the format of 23-25 are different from others.

REPLY: Sincere thanks to the excellent job of reviewers and we modified them.

Reviewer: 2

Comments to the Author(s)

By establish a Stackelberg model, this paper shows players acting as leaders in turn yields corporation. However, I think this paper still has some room for improvement.

1. The formulas are not numbered.

REPLY: In the modified version, equations are numbered.

2. I suggest adding a list of symbolic meanings

REPLY: IN the modified version, the symbols are explained in Table 1 as follows

Table 1 Nomenclature.

$I = \{1, 2\}$	Two players
$S = \{A, B\}$	Strategy space
$a_1(b_1)$	Payoffs of the first (second) player under strategy (A, A)
$a_2(b_2)$	Payoffs of the first (second) player under strategy (A, B)
$a_3(b_3)$	Payoffs of the first (second) player under strategy (B, A)
$a_4(b_4)$	Payoffs of the first (second) player under strategy (B, B)

3. Can this paper analyze the equilibrium under the infinite strategy?

REPLY: This is a very important issue. Actually, Section 3 in this article focuses on infinite strategy.

4. Some language expressions need to be improved.

REPLY: In the modified version, this article is carefully checked.

Sincere thanks to two anonymous reviews and editor again. The modified part is marked in "BLUE".